# Predictors of Patients’ Satisfaction with Primary Health Care Services in the Kingdom of Saudi Arabia: A Systematic Review

**DOI:** 10.3390/healthcare11222973

**Published:** 2023-11-16

**Authors:** Abdullah M. Alshahrani

**Affiliations:** Department of Family Medicine, College of Medicine, University of Bisha, Bisha 67714, Saudi Arabia; abmohammed@ub.edu.sa

**Keywords:** Saudi Arabia, primary care, patients’ satisfaction, systematic review

## Abstract

Background: Understanding the factors influencing patients’ satisfaction with primary healthcare services in the Kingdom of Saudi Arabia is essential for improving healthcare outcomes and patient experiences. Objectives: This research work is concerned with the identification of the patient satisfaction predictors with the primary health care services by conducting a systematic review in the Kingdom of Saudi Arabia. Methods: The three databases in the form of Google Scholar, PubMed, and Medline have been used for article extraction. Keywords have been used to search the articles related to this work, such as the predictors of patient satisfaction. The different journals selected were associated with the selected data basis. The research studies selected for the systematic review were evaluated with the help of PRISMA and JBI assessments. The cross-sectional studies have been included in this systematic review. Results: The 3125 articles identified were from the three databases PubMed (1352), Medline (1103), and Google Scholar (670). All the selected studies were evaluated and screened with the help of PRISMA. After extracting the 25 articles for the systematic review, the JBI assessment was applied to the methodologies. The overall quality satisfaction indicated that all the selected studies were suitable for the systematic review. Conclusion: Studies have consistently identified five key predictors of patient satisfaction in primary healthcare: availability, accessibility, communication, rational conduct, technical skills, and personal qualities. Policymakers, healthcare providers, and researchers can use these insights to inform strategies to optimize healthcare services and foster higher levels of patient satisfaction in the Kingdom.

## 1. Introduction

During the assessment of the contemporary system of health care, patient satisfaction (PS) has garnered immense attention, making it a prime concern of health care professionals and service providers. Established in 1926, the Ministry of Health (MoH) in Saudi Arabia (KSA) maintains established healthcare service quality standards nationwide. MoH network provides healthcare services in the KSA region at three levels: tertiary, secondary, and primary [1].

Primary care lies in the category of first-level care that patients receive, which further includes screening, treatment, health promotion, and disease prevention [2]. Primary healthcare center (PHC) often fulfills most of the needs related to a patient’s health conditions during their lifetime. These needs can be connected to the patient’s social, physical, or mental well-being. It refers to the initial contact with the health care system when they face any illness or health issue and is the first level of medical care [3]. Thus, the participation of the public, cooperation within sectors, promotion of health, adequate technology, and accessibility are the core principles of PHC [4].

PHC has gained attention as a significant healthcare issue in KSA in recent years. This attention has resulted in improvements and advancements in the performance and services of the healthcare sector. More than 2200 facilities for PHC are situated in KSA, which offer high-quality medical services and PHC to expatriates and nationals. The PS is one of the decisive components of the quality evaluation process in the healthcare sector [5]. The link between the expectations regarding health care services and the perceptions about the needs of the patients depicts a complex and detailed phenomenon referred to as the PS. Thus, one of the key variables that significantly influence the performance and quality of the services offered in the healthcare sector is the satisfaction level of the patients [1].

Identifying and eradicating those factors contributing to the dissatisfied attitude of the patients is necessary to improve the inclusive provision and service quality of healthcare facilities. Demand for health care services has increased massively in the KSA region due to the sudden surge in urbanization, leading to the social and economic transformation of the region [6]. As a result, it became essential to evaluate the quality of healthcare services in the area by assessing patient satisfaction (PS) levels. These assessments will facilitate the medical practitioners and institutes in understanding patients’ perspectives, allowing them to improve their services further. Moreover, it can also help in measuring the impact of PS on the decision of patients to follow a particular treatment [7]. Patient satisfaction is one of the most crucial determinants when evaluating health outcomes and the quality of services offered by any healthcare facility. This work aimed to determine the levels and predictors of patients’ satisfaction with primary health care services in the Kingdom of Saudi Arabia.

## 2. Materials and Methods

The systematic review is used in this research work. For this purpose, the author reviewed all the included articles to confirm that all are cross-sectional studies, meet the selection criteria, and assess the studies’ quality. The selected studies are assessed using the PRISMA analysis, and it is useful to extract the most suitable research works that help to find the relationship between patient satisfaction and the five major predictors (Table 1). The systematic review is based on 25 studies selected based on the PRISMA and the JBI assessment.

Data Sources and Search Strategy: the three search engines, namely Google Scholar, PubMed, and Medline, have been used to extract the articles. The keywords have been used to search the articles related to this work, such as the predictors, patient satisfaction, and PHC. The different journals were selected that were associated with the selected data basis. The English language is used to perform the search for required studies. Similar terms, keywords, and Boolean expressions were used to search the articles on these three databases, and the relevant articles have been used to perform the forward citation to explore the studies included in this review based on the most pertinent research works.

In this work, the most relevant search terms used are: “patient satisfaction”, “primary health care sector of Saudi Arabia”, “predictor of patient satisfaction”, and the “Kingdom of Saudi Arabia”. From the search results, 1352 articles were extracted from PubMed, 1103 from Medline, and 670 from Google Scholar. All the journal articles related to healthcare research works are included in extracting the relevant research studies conducted by the previous researchers. All these three databases identified an extensive range of articles related to literature. The major reason behind selecting these three databases is that these databases are more effective, efficient, and used more often than the other databases [8].

Study selection: all the studies chosen for the systematic review underwent evaluations using the PRISMA and JBI assessment criteria. The studies included in this work meet the requirements: the research must be original, conducted between 2007 and 2022, the research area must be KSA, the research determines the patient’s satisfaction with the primary healthcare services, and the research work is based on a cross-sectional study design methodology. The studies were included and excluded based on the results of the JBI assessment and PRISMA [9].

Quality Appraisal (JBI Assessment and setting Domains): Assessment is one of the major critical steps associated with the systematic review. The major aim of the JBI assessment and the domain is to assess the quality and methodology used to determine the possibility of bias based on design, conduct, and analysis [10]. The scientific committee of JBI has officially endorsed using the JBI assessment to establish the inclusion criteria for studies in the systematic review [11].

Data Extraction: the table below (Table 2) consists of the cross-sectional checklist under the JBI assessment used to assess the methodology of the selected studies. These studies are used in the systematic review.

The eight questions have been included in the checklist because the eight questions were included in the cross-sectional checklist associated with the JBI assessment. The responses have been recorded as not applicable, no, yes, and unclear. The questions related to the JBI cross-sectional appraisal have been presented below.

Q1: Were the inclusion criteria for the sample clearly specified? 

Q2: Were the study participants and the research setting thoroughly described? 

Q3: Was the measurement of exposure conducted with validity and reliability? 

Q4: Were standardized, objective criteria employed to measure the condition? 

Q5: Were potential confounding factors identified? 

Q6: Were strategies outlined for addressing confounding factors? 

Q7: Were the outcomes assessed with validity and reliability? 

Q8: Was appropriate statistical analysis applied?

## 3. Results

### 3.1. Data Search

The three databases were used to search more than three thousand articles related to the topic of the study. The 3125 articles have been identified from the three databases with a composition of PubMed (1352), Medline (1103), and Google Scholar (670). Two-hundred fifteen articles were excluded because of the problem of duplication, and after the process of screening, 50 articles were selected after the elimination of the 2860 articles. After that, full-text articles were assessed, and a further 20 articles were excluded for a number of reasons, which include the outcomes being not relevant, out of scope, or insufficient study design description. At the final stage of the PRISMA, the 25 articles were finalized for the systematic review. The results of the preferred reporting items for the meta-analysis and systematic reviews have been presented in Figure 1.

### 3.2. Characteristics of the Included Papers

All the studies included in this analysis originate from Saudi Arabia, as this study is centered around a systematic review of research explicitly conducted within the context of Saudi Arabia. The cross-sectional studies have been included because they are based on originality in their data analysis. The studies have been compared because of the two major approaches in direct and indirect applications of the patient satisfaction domains. The selected cross-sectional studies have been based on the survey because all these studies used primary data. The predictors of patient satisfaction have been identified in this work and presented below. 

### 3.3. Overall Satisfaction

The overall satisfaction of the studies has been measured with the help of the JBI assessment, and it is also based on the questions asked of the participants at the end of the patient satisfaction questionnaire. All the included studies have been based on the overall satisfaction measured, and most of the studies ranged greater than the minimum level, that is, 50 percent, because most of the studies have an overall satisfaction level of more than 80 percent. The overall satisfaction associated with the selected studies is presented below (Table 3). 

These are the five major predictors identified by the various studies in their cross-sectional research because identifying the major predictor of patient satisfaction in the healthcare sector is the major reason behind conducting the research. The results mentioned below (Table 4) indicated that various studies have been showing these five major factors that affect patients’ satisfaction that have been used as the domains of patient satisfaction in the primary healthcare sector [7]. Optimizing patient outcomes seems to be a sensible strategy for enhancing results. When patients are content with their physician and the quality of care they receive, they are more inclined to comply with their treatment regimens. However, a patient might have a great experience and still obtain an unfavorable diagnosis, such as cancer. Patient happiness should be used by healthcare systems as a balancing metric, not as a driver of results. Balanced measures enable health systems to achieve substantial quality-of-care gains despite losing track of possible negative consequences [12].

By using patients’ experience as a balancing metric, healthcare organizations may better ensure that changes in one area will not negatively influence other areas. For instance, the length of stay is the outcome metric for a health system’s effort to cut down on the amount of time patients spend in labor and delivery. However, if mothers have the impression that they are being hurried toward release, reducing the length of stay will harm the patient experience [13]. 

Patient satisfaction is an essential indicator of healthcare quality because it provides information on the provider’s performance in satisfying the client’s expectations, which are of the highest significance to the patient and are a major predictor of patients’ prospective performance expectancy [15]. There is a correlation between patient satisfaction and critical outcomes, such as increased compliance, lower consumption of medical services, fewer instances of a lawsuit for medical negligence, and an improved prognosis. Because of the absence of a strong theoretical framework and a standardized set of measures for consumer satisfaction, surveys have exclusively emphasized patient experiences over the last ten years [16]. These surveys look at aspects of patient care such as long waits, the excellence of basic amenities, and interaction with healthcare professionals, all of which help identify tangible priorities for improving quality care [17].

Positive patient experiences are an essential aim. Several components of patient satisfaction, such as provider-patient solid communication, are linked to crucial healthcare procedures and results. These include patient compliance with medical recommendations, better clinical results, more significant patient safety measures, and less wasteful healthcare use [19]. Although some research finds no connection between the patient’s perspective, clinical procedures, and results, this finding should not come as a surprise. Various elements might influence processes and results in addition to the patient’s experience [20]. When generating a holistic picture of performance, combining patient experience measurements with other quality metrics is essential [21].

There is a distinction between patient satisfaction and patient experience, even though the two phrases are often used indiscriminately with one another. To evaluate patient experience, it is necessary to inquire about the patient’s estimation of whether something that ought to occur in a healthcare context did occur and the frequency with which it did occur [22]. On the other side, what determines a patient’s level of satisfaction is whether their anticipation about a health visit was fulfilled. Two persons can provide different satisfaction rates for the same treatment if they have distinct assumptions about how the care is meant to be given. This is because both individuals received identical care but had different expectations [23].

It is vital to provide excellent service to patients if one is interested in achieving the highest levels of patient satisfaction. It is important to keep in mind that patients have a high level of awareness about certain medical issues, and it is critical to have a patient-friendly approach and follow the institution’s best practices [25]. In addition, the medical professionals, and the rest of the PHC personnel need to be supportive of the patients, and the patients themselves need to feel respected and comfortable enough to have a good impact, not only on their health, but also on how they think about the PHC. They should be supplied with services of the highest possible quality and a sanitary environment [27,30].

## 4. Discussion

One of the objectives of this study was to identify and assess potential predictors of satisfaction, particularly in light of the proliferation of patient satisfaction questionnaires in recent decades as instruments for gauging healthcare quality from the patient’s viewpoint [1,4]. A shared focus group comprises individuals admitted to a hospital, as the admission process can be a source of stress and dissatisfaction for many due to the significant healthcare expenses associated with admission to a healthcare facility. However, most patients require PHC services more frequently than hospital care, so we focus on these people. Like other assessment tools, researchers must evaluate patient satisfaction questionnaires for validity and reliability [34]. These are fundamental characteristics that researchers aim to establish for their instruments. Questionnaires can be completed by self-reporting, face-to-face interviewing, phone interviewing, or, most recently, by computer. In the self-reporting method, the questionnaire is given to the patient at a specific time, either personally, by mail, or by the Internet. However, the Internet is becoming the most frequent method [31]. A major problem and source of bias is the patients who do not complete the questionnaire. Researchers typically send reminders up to three times after the first contact to minimize the number of missing people. Additionally, those who do not respond may be contacted by phone to encourage them to answer the questionnaire. However, this is an additional source of bias that has already been studied [29]. Hospitalized patients are typically of advanced age, and in certain instances, they may face functional limitations that hinder their ability to fill out a questionnaire independently. In such situations, patients may rely on assistance from a family member or friend to respond to the questionnaire, potentially introducing a source of bias. In 2002, a validated inpatient satisfaction questionnaire was used to evaluate the health care received by patients admitted to several hospitals [34]. Distinguishing from other questionnaires, this assessment segmented the data into separate domains, generating individual scores to facilitate analysis. They opted for a self-reported questionnaire distributed via mail, permitting patients to complete it independently or with the aid of a family member or friend, provided they indicated the helper’s identity. These are essential aspects that researchers aim to demonstrate for their instruments.

Additionally, it is important to acknowledge that other potential sources of bias may emerge during the data collection and subsequent analysis phases. Patient satisfaction varies from country to country and even among nations with comparable health outcomes and healthcare infrastructure. The patient experience has been shown to explain 10% of the diversity in patient satisfaction across nations [19]. Surveys are administered at public and commercial healthcare institutions to investigate the patient experience and the variables that influence it.

Moreover, one of the surveys that was carried out at four of the most important public hospitals in Saudi Arabia revealed that the structure of the hospital, evaluated in terms of the accessibility of medical care, the building, the cleanliness of the rooms, and the number of available beds, affects the degree to which patients are satisfied [12]. Patient dissatisfaction was caused mainly by a lack of basic amenities, a shortage of doctors and paramedical staff, a lack of beds, long wait times to get admitted into the hospital, a lack of doctors and paramedical staff, and a problem with sanitation. The level of formal attention paid to the definition of the idea has been lacking despite the level of satisfaction being a measure of the quality of treatment offered to patients [13,33].

Upon reviewing the available literature, it can be inferred that patients tend to experience higher satisfaction with healthcare services when the healthcare system exhibits responsiveness in areas, such as respecting their dignity and autonomy, offering timely attention, and meeting their expectations for satisfaction. Several studies discovered that patient expectations, which are impacted by patient factors such as age, socioeconomic class, education, and, to a lesser degree, gender, and ethnicity, are significant determinants of patient satisfaction. Nonetheless, patient views and other psychological aspects may be disregarded as a role [14]. A few studies in Pakistan reveal that the commercial healthcare industry is somewhat responsive; however, the state sector is grossly underused, and there is no idea of quality improvement or high-quality service supply in government institutions [15]. Capacity building of health professionals, particularly the training of health employees in communication and interpersonal skills, is one of the accessible and practical ways to increase patient satisfaction. Most patient satisfaction surveys support this observation and may be more appropriate for resource-poor nations because it is more cost-effective than developing technical facilities [16]. 

The assessment of patients’ perceptions regarding their duration of stay in healthcare facilities is emerging as an increasingly significant metric for evaluating the quality of care provided by the healthcare industry in contemporary times. Concerning the topic of patient satisfaction, several aspects, such as the behavior of the staff, the contact between patients and physicians, concerns over the management of the health facility, and the physical environment, are essential components. In addition to being a barometer of patient outcome, patient happiness is also a barometer of other health indicators of an institution [35]. A patient who is happy with the therapy they are receiving is more likely to follow the prescribed regimen and return for frequent checkups. Therefore, it is of the highest significance to be aware of patient expectations and the degree to which they are satisfied to provide high-quality medical treatment [18].

Nearly all the patient satisfaction surveys carried out globally are designed to assess the level of satisfaction a patient has with their healthcare provider to improve the level of treatment provided. As an indication of how responsive a healthcare system is, the World Health Organization employs metrics that quantify the experience that patients have with the healthcare system. The system’s performance and responsiveness are demonstrated by the overall enhancement of the health status of its beneficiaries. This improvement ensures equality and efficiency while shielding individuals from catastrophic costs [24,28,36]. According to the World Health Organization (WHO), one of the best ways to evaluate how responsive a healthcare system is would be to poll the general population on their experiences using various medical services. The health care system’s responsiveness is directly linked to patient satisfaction, the quality of health care provided, and the patient’s personal experience. When individuals seek medical attention, they should expect it to be handled in a way that is sensitive to their needs and the setting where they receive treatment [37].

A person’s level of contentment with a healthcare institution may be deduced from their level of contentment with the organization on numerous fronts. The degree to which a patient’s expectations of the services and care they receive are met by the patient’s perceptions of the services and care provided is one way to define satisfaction with the quality of health care provision [26,32]. Individuals who do not fill out the questionnaire represent a significant concern and a potential source of bias. After the first letter, researchers will often send reminders, perhaps as many as two or three of them, to reduce the number of persons who have gone missing. In addition, they may attempt to persuade individuals who do not reply to the questionnaire by calling them and pleading with them to do so, even though this is an additional form of bias that has previously been investigated [38].

## 5. Conclusions

Through a systematic review, this research focuses on identifying the factors that predict patient satisfaction with primary healthcare services in the Kingdom of Saudi Arabia. 

The five domains used for patient satisfaction include availability and accessibility, personal qualities and attributes related to communication, technical skills, and rational conduct. The PRISMA is used to identify the new studies through databases, which include PubMed (1352), Medline (1103), and Google Scholar (670). After screening the studies, 50 articles were extracted, and five were extracted further. The 45 full-text articles were assessed for eligibility. Twenty articles were excluded for several reasons, including the outcomes being irrelevant, out of scope, or having insufficient study design description, and 25 studies were included in the systematic review. Most studies included the five major predictors of patient satisfaction in primary healthcare, which are availability and accessibility, communication-related attributes, rational conduct, and technical skills, along with personal qualities.

### Recommendation

This study concludes that the satisfaction of patients must be the primary goal of the primary healthcare sector, considering each of the factors of patient satisfaction that have been identified in this work. The communication attribute in the form of listening skills and ensuring the understanding of the patient should be improved. Relational conduct must be enhanced by treating the patient respectfully, and patient trust and confidence should also improve. The technical skills and personal qualities should be enhanced by including the knowledge and expertise of the professionals. The accessibility and availability must be improved in the primary healthcare sector of Saudi Arabia regarding patient satisfaction.

## Figures and Tables

**Figure 1 healthcare-11-02973-f001:**
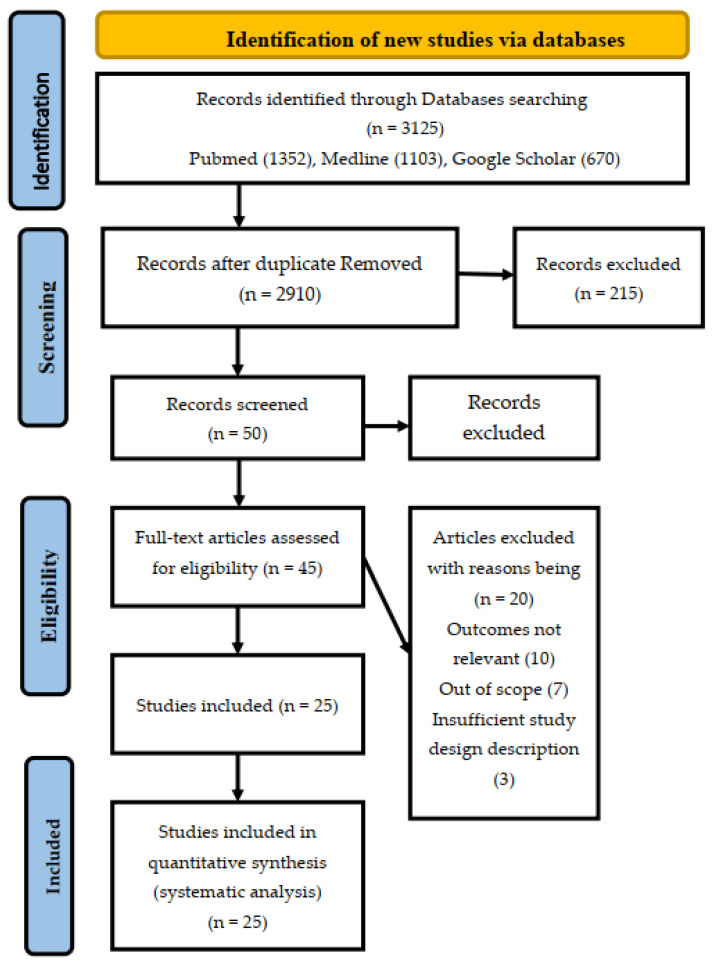
PRISMA flow Diagram for Study Selection for Systematic Review.

**Table 1 healthcare-11-02973-t001:** The domains of patient satisfaction with doctors’ measurement.

Domain	Patient-Physician Measurement
Accessibility and availability	The physician was accessible. The patient spent sufficient time with the physician, and the patient felt good.
Personal qualities	Kindness, empathy, concern, sensitivity, and friendliness (humaneness)
Attributes related to communication	Addressing the patient’s questions and concerns Ensuring patient understanding Providing the explanation Listening skills Providing the information Eliciting patient information
Relational conduct	Patients feel confident in the healthcare Health problems are taken seriously The patient should be treated respectfully Allowed the patient a shared role in the decision-making and medical care Patient felt understood Patient felt heard Professional demeanor
Technical skill	Professional expertise and knowledge

**Table 2 healthcare-11-02973-t002:** Results of critical appraisal results for included studies using the JBI cross-sectional critical appraisal checklist.

Citations	Q1	Q2	Q3	Q4	Q5	Q6	Q7	Q8
Almoajel et al. (2014) [7]	Y	Y	NA	Y	Y	Y	U	Y
Owaidh et al. (2018) [12]	Y	N	Y	Y	NA	Y	U	Y
Abolfotouh et al. (2017) [13]	Y	NA	Y	Y	Y	N	Y	Y
Makeen et al. (2020) [14]	Y	U	Y	Y	NA	Y	Y	Y
Al-Ali et al. (2020) [15]	Y	Y	NA	Y	Y	N	Y	NA
Alfaqeeh et al. (2017) [16]	Y	NA	Y	N	Y	Y	NA	Y
Bawakid et al. (2017) [17]	Y	NA	Y	Y	Y	N	Y	Y
Mohamed et al. (2017) [18]	Y	N	Y	Y	Y	NA	Y	Y
Almezaal EA et al. (2021) [19]	Y	Y	NA	Y	Y	Y	N	Y
Almutairi (2017) [20]	Y	Y	NA	Y	Y	Y	Y	Y
Al-Makhaita and Sabra (2014) [21]	Y	N	Y	Y	Y	U	Y	Y
Alosaimi et al. (2022) [22]	Y	Y	Y	NA	Y	Y	Y	Y
Senitan and Gillespie (2019) [23]	Y	Y	NA	Y	Y	Y	U	Y
Elias et al. (2022) [24]	Y	N	Y	Y	Y	Y	NA	U
Ahmed et al. (2022) [2]	NA	Y	Y	Y	Y	Y	Y	Y
Alsayali et al. (2019) [25]	Y	Y	Y	Y	Y	U	Y	Y
AlOmar et al. (2021) [26]	Y	Y	NA	Y	Y	Y	N	Y
Llego and Al-Shirah (2017) [27]	Y	Y	Y	NA	Y	N	Y	Y
Sadovoy et al. (2017) [28]	Y	Y	NA	Y	Y	U	Y	UA
Alturki and Khan (2013) [29]	Y	Y	NA	Y	U	Y	NA	Y
Albahrani (2022) [30]	Y	U	Y	Y	Y	NA	Y	Y
Alzaid et al. (2016) [31]	Y	Y	NA	Y	N	Y	U	Y
Mohamed et al. (2015) [32]	Y	N	Y	Y	U	Y	NA	Y
Alrasheedi and Al-Mohaithef (2019) [33]	Y	Y	U	Y	N	Y	NA	Y
Alotaibi et al. (2021) [6]	Y	N	Y	NA	Y	Y	U	Y

Y (Yes); N (No); U (Unclear); NA (Not Applicable).

**Table 3 healthcare-11-02973-t003:** The overall satisfaction.

Study	Overall Stratification %
Almoajel et al. (2014) [7]	78
Owaidh et al. (2018) [12]	80
Abolfotouh et al. (2017) [13]	79
Makeen et al. (2020) [14]	88
Al-Ali et al. (2020) [15]	82
Alfaqeeh et al. (2017) [16]	85
Bawakid et al. (2017) [17]	90
Mohamed et al. (2017) [18]	95
Almezaal EA et al. (2021) [19]	85
Almutairi (2017) [20]	78
Al-Makhaita et al. (2014) [21]	80
Alosaimi et al. (2022) [22]	90
Senitan and Gillespie (2020) [23]	95
Elias et al. (2022) [24]	80
Ahmed et al. (2016) [2]	89
Alsayali et al. (2019) [25]	86
AlOmar et al. (2021) [26]	82
Llego and Al-Shirah (2017) [27]	85
Sadovoy et al. (2017) [28]	90
Alturki and Khan (2013) [29]	95
Albahrani et al. (2022) [30]	96
Alzaid et al. (2016) [31]	89
Mohamed et al. (2015) [32]	88
Alrasheedi and Al-Mohaithef (2019) [33]	85
Alotaibi et al. (2021) [6]	80

**Table 4 healthcare-11-02973-t004:** Domains related to predictor of patient satisfaction with the primary health care services.

Domain Article	Communication Attributes	Rational Conduct	Technical Skill and Knowledge	Personal Qualities	Availability and Accessibility	Total of Predictors
Almoajel et al. (2014) [7]	Y	Y	Y	N	Y	4
Owaidh et al. (2018) [12]	N	N	Y	Y	N	2
Abolfotouh et al. (2017) [13]	Y	Y	N	N	Y	3
Makeen et al. (2020) [14]	N	Y	Y	Y	N	3
Al-Ali et al. (2020) [15]	Y	N	N	N	Y	2
Alfaqeeh et al. (2017) [16]	Y	Y	N	Y	Y	4
Bawakid et al. (2017) [17]	Y	N	Y	N	N	2
Mohamed et al. (2017) [18]	N	N	Y	Y	Y	3
Almezaal et al. (2021) [19]	Y	N	Y	N	Y	3
Almutairi (2016) [20]	Y	Y	N	N	N	2
Al-Makhaita et al. (2014) [21]	N	Y	Y	Y	N	3
Alosaimi et al. (2022) [22]	N	Y	N	Y	Y	3
Senitan and Gillespie (2019) [23]	N	Y	Y	N	Y	3
Elias et al. (2022) [24]	N	Y	N	Y	Y	3
Ahmed et al. (2016) [2]	N	N	Y	Y	N	3
Alsayali et al. (2019) [25]	Y	Y	N	N	Y	3
AlOmar et al. (2021) [26]	N	N	Y	Y	N	2
Llego and Al-Shirah (2017) [27]	Y	Y	N	N	Y	3
Sadovoy et al. (2017) [28]	N	Y	N	Y	Y	3
Alturki and Khan (2013) [29]	Y	N	Y	N	Y	3
Albahrani et al. (2022) [30]	Y	Y	N	Y	N	3
Alzaid et al. (2016) [31]	N	N	Y	Y	N	3
Mohamed et al. (2015) [32]	N	N	Y	Y	Y	3
Alrasheedi and Al-Mohaithef (2019) [33]	Y	N	N	Y	N	2
Alotaibi et al. (2021) [6]	N	Y	Y	N	Y	3

## Data Availability

Not applicable.

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
