# Peer review of "Predictors of Patients’ Satisfaction with Primary Health Care Services in the Kingdom of Saudi Arabia: A Systematic Review"

_healthcare, 2023, doi:10.3390/healthcare11222973_

Round 1

Reviewer 1 Report

Comments and Suggestions for Authors

Thank you for the opportunity to review the article "Predictors of Patients’ Satisfaction with Primary Health Care Services at the Kingdom of Saudi Arabia: A Systematic Review."

This study provides valuable insights into improving patient satisfaction in the primary healthcare sector of Saudi Arabia. The article makes a significant contribution to the field of healthcare improvement, emphasizing the pivotal role of the patient in the healthcare process. The authors present compelling arguments for a patient-centered approach, demonstrating that it is not only ethically sound but also leads to tangible improvements in the quality of medical services. Based on the identified factors, the authors formulated recommendations and specific areas in need of improvement:

1. Healthcare professionals should focus on enhancing their communication skills, particularly listening skills and ensuring that patients fully understand the information provided. Effective communication is key to patient satisfaction.

2. Treating patients with respect and ensuring they feel heard and understood is essential. Building patient trust and confidence in healthcare providers should be a priority.

3. Healthcare professionals should continuously update and improve their technical skills and knowledge. This will contribute to a higher level of patient satisfaction.

4. Personal qualities of healthcare professionals, such as kindness, empathy, and friendliness, play a significant role in patient satisfaction. Healthcare facilities should prioritize these attributes in their staff.

5. Improving the accessibility and availability of healthcare services in the primary healthcare sector. Reducing wait times, addressing the shortage of medical staff, and ensuring basic amenities are readily available can lead to increased patient satisfaction.

The conclusions and recommendations from the study underscore the importance of patient satisfaction as the primary goal of the primary healthcare sector in Saudi Arabia. The summary and presentation of recommendations are precise and well-constructed. Implementing these recommendations in the primary healthcare sector in Saudi Arabia can significantly enhance the quality of care, patient satisfaction, and overall healthcare outcomes. It also highlights the significance of continuous professional develo pment and the role of healthcare institutions in creating a patient-friendly environment.

My information is as follows:

- line 67 in the author's description, with detailed access to all articles and additional confirmation of their quality - does he have access to the authorization?

- define the decision-making process based on the indication of a specific group of members;

- line 217 in which an indication of an inpatient care physician is found, and the study is about a care physician

- line 272 Application studies - how valuable? - Verse 279 barometer of other indicators of institutional performance that can be supplemented?

Author Response

Please see the attachment (point-by-point response to the reviewer’s comments)

Reviewer 2 Report

Comments and Suggestions for Authors

a valuable publication from the point of view of Proms & Prems. At PHC we are still waiting for indicators that will allow mothers to assess the patient's approach. it would be interesting to compare with other countries in the next steps. the analysis process was performed using appropriate assessment tools and methods.

Author Response

(The authors gave the same response as above.)

Reviewer 3 Report

Comments and Suggestions for Authors

Dear Authors,

I appreciated your paper. Please find my comments below:

1. the methodology of systematic review is sound and well described. Anyway, can you say why you consider sufficient 25 paper selected? is because they represent different approaches or different contexts?

2. Does the paper you analyzed consider the same evaluation scale 

3. CAn you explicit and strengthen the relations among expexctations, quality of treatments satisfaction by patients? for example if the expectation are higher the satisfaction could be lower even if the quality of services is the same.

4. For an international reader, it wiould be helpful the description of the primary care in Saudi Arabia. The aim is to contextualize the analisys. In the conclusions,  you repeat the aspect of the methodology you slready described in the prevos paragaph. is it redundant?

5. In order to strengthen the systematic review you can say something about the implication f your analisys, I mean do you think that it will be necessary to update the training processes of the heath professionals (primary care nurses, doctors etc.). Or do you see messages for managers.

6. Last but not least, do you think that the cultural aspects are relevant? can you say something about the culture. is it possible to say something about the relations between expectation and satisfaction from one hand and actual quality of primary care services and satisfaction

Author Response

(The authors gave the same response as above.)
